# Data-Driven Discovery of Functional Cell Types that Improve Models of Neural Activity

**Daniel J. Zdeblick**[*]     **Eric T. Shea-Brown**[† ‡]     **Daniela M. Witten**[§]     **Michael A. Buice**[‡ †]

## Abstract

Computational neuroscience aims to fit reliable models of in vivo neural activity and interpret them as abstract computations. Recent work has shown that functional diversity of neurons may be limited to that of relatively few cell types; other work has shown that incorporating constraints into artificial neural networks (ANNs) can improve their ability to mimic neural data. Here we develop an algorithm that takes as input recordings of neural activity and returns clusters of neurons by cell type and models of neural activity constrained by these clusters. The resulting models are both more predictive and more interpretable, revealing the contributions of functional cell types to neural computation and ultimately informing the design of future ANNs.

## 1 Introduction

The primary goal in the field of computational neuroscience is to build mathematical models that link the in vivo activity of our brains with the intelligent behavior they produce. The primary obstacles to accomplishing this stem from the brain's complexity. The large number and variable characteristics of individual neurons in a given brain produce highly nonlinear and high dimensional activity. This makes theoretical analysis difficult and limits our observations of the brain to a subset of neurons, making computational models fitted to data less reliable.

Luckily, applying biological constraints to our models may help us understand the brain. The ANN trained to perform spatial localization by Cueva [3] required a metabolic constraint on total neural activity to reproduce patterns of neural responses in the Entorhinal Cortex. The ANN trained by Yamins [9] to recognize objects required a convolutional structural constraint to reproduce activity of the visual system. These works have strengthened the connection between AI and neuroscience by showing how the same constraints produce similar activity and behavior.

In this work, we seek to make this connection with a different constraint, cell diversity limited to small variations within relatively few cell types. The Allen Institute for Brain Science has pioneered projects for identifying such cell types by looking at transcriptomic, morphological, and electrophyisiological features of individual neurons [5], [6]. This idea of discrete cell types is also in line with theoretical results analyzing how simple point models of neural activity undergo bifurcations in parameter space between a few qualitatively different behaviors (see [4] for bifurcation analysis of the Izhikevich model, and [8] for evidence that the GLM has similar stereotypical behaviors). We take a bottom-up approach of inferring functional cell types from neural activity data and using these types to constrain single-cell activity models. By using cell types as a constraint, we can fit more reliable models for individual neurons and ultimately uncover the roles of functionally distinct cell types in computations by real and artificial brains.

---

[*]Electrical and Computer Engineering, University of Washington      [†]Applied Math, University of Washington      [‡]Allen Institute for Brain Science      [§]Statistics, University of Washington

33rd Conference on Neural Information Processing Systems (NeurIPS 2019), Vancouver, Canada.

## 2 Methods

### 2.1 Generative Model

We use a Generalized Linear Model (GLM) to model the single-cell spiking activity $x_i(t)$ in response to a stimulus $s_i(t)$: $x_i(t) \sim Poisson(exp[(s_i \star F_i)(t) + (x_i \star W_i)(t) + b_i]) := \mathcal{D}(s_i(t), W_i, F_i, b_i)$, where $(a \star b)(t) = \int_{-\infty}^{\infty} a(\tau)b(t - \tau)d\tau$, $F_i$ is the stimulus filter for cell $i$, $W_i$ is the self-interaction filter for cell $i$, and $b_i$ is the bias term for cell $i$ (see [7] for a thorough introduction). We hypothesize that $W_i$ will be closely related to the functional type of cell $i$ while $F_i$ and $b_i$ will be more specific to each cell, since the self-interaction filter captures many of the intrinsic dynamical properties that neurons possess [8]. To make our cell types supposition explicit and guide algorithm development, we specify the following generative model for a neural activity dataset, consisting of two stages (see Figure 1 A for a diagram):

1. The self-interaction filter of neuron $i$, $W_i$, is drawn from a $K$-component Gaussian Mixture Model (GMM) where each component corresponds to a cell type:
   $P(W_i|\{\pi_k, \mu_k, \Sigma_k\}) = \sum_{k=1}^{K} \pi_k \mathcal{N}(W_i; \mu_k, \Sigma_k)$. The other cell-specific parameters $F_i, b_i$ are selected from some common (e.g. flat) prior distribution.
2. The cell generates a spiking response $x_i(t)$ to its incoming stimulus $s_i(t)$:
   $x_i(t) \sim Poisson(exp[(s_i \star F_i)(t) + (x_i \star W_i)(t) + b_i])$.

In this work we consider only diagonal covariance matrices $\Sigma_k$, but full covariance matrices could be fit with data from enough neurons $N >> |W_i|^2$

### 2.2 E-M Algorithm

Our goal is to recover the parameters of the generative model from a dataset $s_i(t), x_i(t), i \in \{1, ..., N\}, t \in \{1, ..., T\}$. We use an Expectation-Maximization algorithm to estimate the distribution over the variables latent to each neuron's activity, $W_i$, while computing a point estimate of the global parameters, $\pi_k, \mu_k, \Sigma_k \ \forall k = \{1, ..., K\}$ (see Algorithm 1 for an overview).

The Expectation (E) step estimates the posterior distribution over the latent variables, $P(W_i|\{\pi_k, \mu_k, \Sigma_k\}, s_i(t), x_i(t), F_i, b_i)$. Because $W_i$ are continuous random variables, we use Variational Inference to make estimating this distribution tractable by approximating it with a simple function and minimizing the KL divergence between the approximation and true posterior. Here, we use a multivariate Gaussian with diagonal covariance matrix: $W_i \sim \mathcal{N}(m_i, C_i)$. To minimize the KL divergence between this approximation and the true posterior distribution $P(W_i|\{\pi_k, \mu_k, \Sigma_k\}, s_i(t), x_i(t), F_i, b_i) \ \propto \ \mathcal{D}(x_i(t)|s_i(t), W_i, F_i, b_i)(\sum_{k=1}^{K} \pi_k \mathcal{N}(W_i; \mu_k, \Sigma_k))$, we sample from the posterior and fit $m_i, C_i$ to the samples. To accomplish this, we use importance sampling, drawing samples $W_s$ from the GMM and assigning them likelihood weights,

---

**Data:** $s_i, x_i \ \forall i \in \{1, ..., N\}, K$
**Result:** $\{\pi_k, \mu_k, \Sigma_k\} \ \forall k \in \{1, ..., K\}, m_i, C_i \ \forall i \in \{1, ...N\}$
initialize $W_i, F_i, b_i$ with maximum likelihood parameters for $s_i, x_i$;
initialize $\{\pi_k, \mu_k, \Sigma_k\}$ with GMM fit to $W_i$;
**repeat**
    /* E-Step: approximate distributions over latent variables $W_i$     */
    **for** *each neuron $i$* **do**
        Sample $W_s$ from the GMM parameterized by $(\{\pi_k, \mu_k, \Sigma_k\} \ \forall k) \ \forall s = \{1, ..., S\}$;
        $w_s \leftarrow \mathcal{D}(x_i(t)|s_i(t), W_s, F_i, b_i) \ \forall s \in \{1, ..., S\}$;
        $\tilde{w}_s = \frac{w_s}{\sum_{s'} w'_s} \ \forall s \in \{1, ..., S\}$;
        $m_i \leftarrow \sum_s \tilde{w}_s W_s$;
        $C_i \leftarrow \sum_s \tilde{w}_s (W_s - m_i)^2$;
    **end**
    /* M-Step: find maximum likelihood global parameters $\{\pi_k, \mu_k, \Sigma_k\}$     */
    Sample $W_{i,r} \sim \mathcal{N}(m_i, C_i) \ \forall r = \{1, ..., R\}, \ \forall i = \{1, ..., N\}$;
    $\{\pi_k, \mu_k, \Sigma_k\} \leftarrow$ best fit GMM to $\{W_{i,r}\}$;
**until** *convergence*;

**Algorithm 1:** E-M algorithm

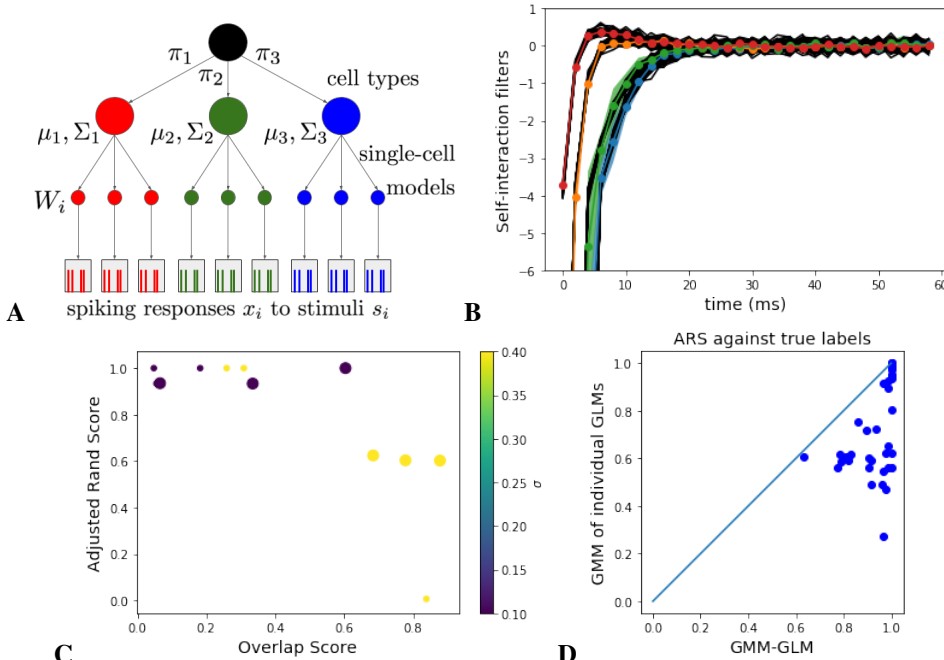

Figure 1: **Simulated Datasets.** A: The generative model described in Section 2.1. B: True self-interaction filters (black) and fitted cluster centers $\mu_k$ (colors) for an example simulated dataset with $K = 4$, $\sigma = 0.1$; shaded region is $\pm\sqrt{diag(\Sigma_k)}$. C: Adjusted Rand Score (ARS) versus Max Overlap Score for 12 simulations of 120 model GLM neurons each. Small circles are for simulations with $K = 2$, large ones for $K = 4$. D: Comparison of our method to fitting a GMM to individually fit GLM parameters; our method discovers classes with a higher ARS to the true labels

$w_s = \mathcal{D}(x_i(t)|s_i(t), W_s, F_i, b_i)$. After these weights are normalized, $\tilde{w}_s = w_s / \sum_{s'} w'_s$, they are used to define the expectations necessary to estimate the moments $m_i \approx \sum_s \tilde{w}_s W_s$, and $C_i \approx \sum_s \tilde{w}_s (W_s - m_i)^2$.

The Maximization (M) step consists of finding $\{\pi_k, \mu_k, \Sigma_k\}$ that minimize the KL divergence between the approximated distribution over $W$, $P(W) \approx 1/N \sum_i \mathcal{N}(W|m_i, C_i)$, and the mixture model. To do this, we sample the approximated distribution $W_{i,r} \sim \mathcal{N}(m_i, C_i) \ \forall r = \{1, ..., R\}$, $\forall i = \{1, ..., N\}$, then fit the mixture model to these samples using standard methods.

## 3 Results

We focus on applying Algorithm 1 to the In Vitro Single Cell Characterization (IVSCC) dataset collected by the Allen Institute for Brain Science [1]. Specifically, we model the spiking response of isolated neurons to a pink noise stimulus.

### 3.1 Simulation Results

First, we demonstrate that the algorithm accurately recovers the true parameters of a generative model from simulated data. We use the same stimuli that were presented in the IVSCC dataset, and simulate responses of GLMs with parameters sampled from a GMM. We vary the number of clusters and the within-cluster variance $\sigma^2$ ($\sigma^2 I = \Sigma_k \ \forall k$), and repeat for several different GMMs. We then use the Adjusted Rand Score to provide an unbiased measure of clustering accuracy. To account for chance variations in the selection of self-interaction filters, results are reported with respect to an Overlap Score, where a score of 1 means that two clusters are indistinguishable, and a score of 0 means all clusters are perfectly separated. Figure 1 shows that the algorithm successfully recovers the latent classes when there is not too much overlap. In particular, Figure 1 D shows that our method outperforms clustering individually fitted parameters.

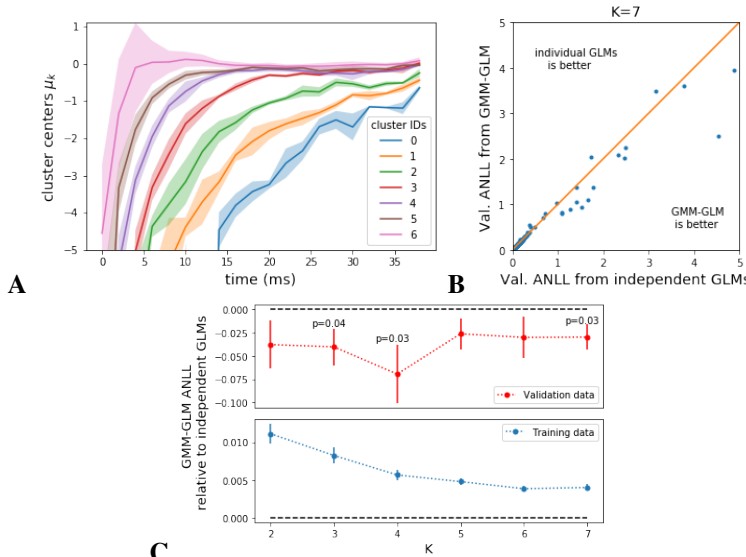

Figure 2: **IVSCC Dataset.** A: Cluster centers $\mu_k$ of self-interaction filters fit to data; shaded region is $\pm\sqrt{diag(\Sigma_k)}$. B: Validation ANLL for each neuron, using the GMM-GLM model versus using independent GLMs. C: ANLL difference between our method and independent GLMs (mean $\pm$ SEM with significant p-values) for the IVSCC dataset.

## 3.2 IVSCC Data

To assess how well the model is able to explain real neural data, we apply Algorithm 1 to spiking data from 290 cells in the IVSCC dataset. As there is no available ground truth for the cell types, we compare the Average Negative Log Likelihood (ANLL) of held out data to that achieved by independently fit GLMs to assess model performance. Figure 2 A-B shows the results of fitting a GMM-GLM model with $K = 7$ components to the dataset, and Figure 2 C compares models with different $K$ according to ANLL. $K = 4$ clusters seem to fit this dataset best, evidenced by a minimum in the validation ANLL and an elbow in the decline of training ANLL with increasing $K$.

With the IVSCC dataset, we are fortunate to have limited morphological, locational, and transcriptomic information available about each cell in addition to the electrophysiological recordings. It is of scientific interest to see if the electrophysiological cell types that we discover are at all related to other partitions of the cells based on these metadata, in the same spirit as [5] and [6]. Note, however, that these metadata do not constitute a "ground truth" for cell types - they merely provide different dimensions along which neurons can be clustered; any similarities (or lack thereof) between our discovered types and the metadata is a scientific result, not a direct validation of our method.

Figure 3 shows the results of comparing these metadata to the $K = 7$ clusters discovered by our method. Evidently, cells that have spiny dendrites, are in cortical layer 1, or express most Cre lines were assigned to certain clusters much more often than chance. The correspondence we find between our clusters and transgenic line is comparable to that found by Teeter et al. (Figure 6 in [6]), and our ARS is slightly higher.

## 4 Discussion

In this work, we make particular choices for the mixture model (GMM), single-cell model (GLM), and which parameters are related to cell type (Self-interaction filters, $W_i$), but our algorithm generalizes to any choices for these. In future work, we will perform model selection over other options.

Ultimately, we seek to apply this algorithm to in vivo neural recordings, so testing the algorithm's robustness to noise is also important. When we apply the algorithm to in vivo data, it may be necessary to use other constraints present in the brain, as well as any available metadata (cell morphology, gene expression, spiking waveform, etc.).

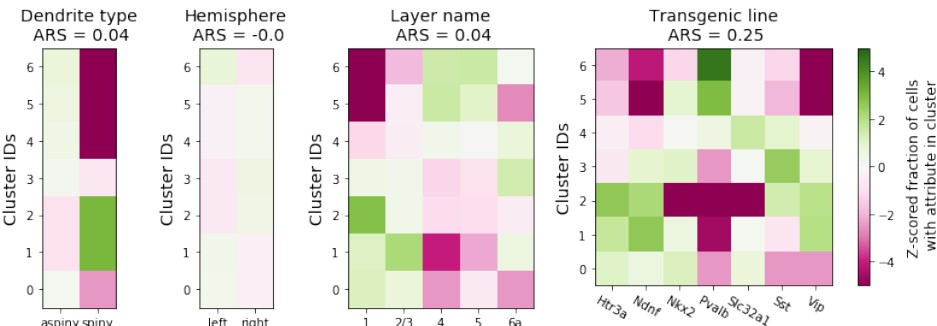

Figure 3: **IVSCC Metadata.** Z-scored differences between the cluster distribution of cells with each attribute and the distribution of all cells. Clusters are the same as in Figure 2 A-B, and attributes are spiny or aspiny dendrites, location (hemisphere and cortical layer), and Cre line. Note the ARS between the cluster and metadata labels in each title. Z-scores are calculated as $Z_i^{(a)} = (\hat{p}_i^{(a)} - \hat{p}_i)/\sqrt{\hat{p}_i^{(a)}(1 - \hat{p}_i^{(a)})/N^{(a)} + \hat{p}_i(1 - \hat{p}_i)/N}$, where $\hat{p}_i$ is the empirical probability that a cell is in cluster $i$ and $\hat{p}_i^{(a)}$ is the empirical probability that a cell with attribute $a$ is in cluster $i$, $N$ is the number of cells, and $N^{(a)}$ is the number of cells with attribute $a$.

Neural Network models with functional cell types that this algorithm produces can support the growing body of theoretical literature regarding such networks, biological and artificial (e.g. [2]). In return, such theoretical techniques can provide a guide for understanding the functional cell type network models our algorithm produces in terms of more abstract network operations.

## 5    Acknowledgements

We would like to acknowledge funding from R01DA047869, and R01EB026908, and a Simons Investigator Award in Mathematical Modeling of Living Systems to Daniela Witten. We thank the Allen Institute founder, Paul G. Allen, for his vision, encouragement, and support.

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
