# OpenReview forum: "Data-Driven Discovery of Functional Cell Types that Improve Models of Neural Activity"
_NeurIPS.cc/2019/Workshop/Neuro_AI — Real Neurons & Hidden Units @ NeurIPS 2019 Poster_

### Official Review · AnonReviewer2 · 2019-09-24
**Interesting approach to studying cell types using electrophysiological data, but very preliminary**

**Clarity:** 3

**Comment:**

I think this work is too preliminary for the moment. Either it needs some results, or it needs to make a more thorough and convincing case as a proof of principle.

**Category:**

Not applicable

**Clarity Comment:**

Quite dense but seems to contain all the relevant information.

**Evaluation:**

2: Poor

**Importance:**

2: Marginally important

**Importance Comment:**

This could lead to important work that allows for a more objective approach to understanding cell types, but for the moment it's just an early proof of principle without any strong results.

**Intersection:**

3: Medium

**Intersection Comment:**

This seems more like a classic neuroscience / statistical approach (E-M algorithm to fit a filtering model to data), but certainly not unrelated to ML.

**Rigor Comment:**

Approach seems reasonable, but I note that the method only seems to work well on simulated data if cell classes are quite well separated, which is probably not the case in real brains.

**Technical Rigor:**

3: Convincing

---

### Official Review · AnonReviewer3 · 2019-09-25
**Potentially interesting method of fitting single-cell models, but method requires further work**

**Clarity:** 3

**Comment:**

A substantial focus of this paper was the fitting of the GMM-GLM, which is somewhat interesting in its own right, but the specific scientific advancement of this technique isn't wholly convincing to me, and therefore I am a little unconvinced that this technique is worth the effort. In particular, I am not convinced that folding in the cell-clustering aspect to the model fitting doesn't do more than regularize the self-interaction filters. Further (and relatedly), I am also not convinced that there is clustering in the self-interaction filters of the neural data (Figure 2B looks like lines were drawn in the sand, although this could be a result from the PCA projection). Overall, it seems that almost the same amount of scientific inference can occur if individual GLMS are fit, and then the interaction filters are clustered afterwards. What do we gain by doing it the way presented (which feels much more complicated)?

In addition, this may be a point of semantics, but functionally-defined cell types are often delineated by the functional properties of the neurons - eg their responses to stimuli. (Or at least often enough that I think the language used in the paper should be clarified.) The self-interaction term based method of classifying feels more like a proxy to classifying cells based on their electrophysiological properties (e.g. fast-spiking interneurons versus excitatory neurons). I think this way of classifying is fine and interesting, but the language used by the authors, and how they relate it to other literature, feels a bit confusing to me.

**Category:**

Neuro->AI

**Clarity Comment:**

The introduction/motivation section was relatively clearly-written. I found section 2.2 to be a bit dense and difficult to read (and I believe that not all of the symbols used were defined). The figures were relatively well-described and easy to interpret.

**Evaluation:**

2: Poor

**Importance:**

2: Marginally important

**Importance Comment:**

Taking the idea that there are multiple, discrete cell types into the fitting of GLMs of these models is an interesting and important idea. Frequently, the investigation of discrete cell types is taken into account after fitting models (typically to see if there is any evidence of discretization), and I think there could be something to gain from using this feature as a prior. However, I was not fully convinced that the results presented in this paper made substantial gains on this question.

**Intersection:**

3: Medium

**Intersection Comment:**

This paper focuses mostly on techniques to fit GMM-GLMs, and feels mostly like an intersection between ML and neuroscience. I think that intersection is very interesting and fruitful, even though it might not be "AI" and neuroscience specifically.

**Rigor Comment:**

The work presented - the GMM-GLMs and the methods used to fit these models - seemed quite rigorous. I appreciate that the authors included details in their algorithm and fitting procedure, but this section was a bit dense to read (which might be necessary but I also wish that some of this text was exchanged for a focus on the specific advancement of this work in understanding neural data). The  testing of the model-fitting on simulated data is very key here.

**Technical Rigor:**

3: Convincing

---

### Official Review · AnonReviewer1 · 2019-09-27
**Interesting constraint in the GLM, but could use validation.**

**Clarity:** 4

**Comment:**

Although validation likelihoods are shown, there should be more validation that the clustering correlates with some other properties of the neurons, ex. some aspect of the morphology.

**Category:**

AI->Neuro

**Clarity Comment:**

Well motivated. Model was well described. The validation needs work, as the clusters have no meaning yet- but this can be performed using the same dataset - I would urge the authors to do so.
Figure 1c was unclear. Figure 1b also is a little unclear - are the self interaction filter cluster centers very close to each other? Please zoom in for the last part of the graph to make out differences in the centers.

**Evaluation:**

4: Very good

**Importance:**

3: Important

**Importance Comment:**

This kind of model is interesting to classify cells based on their dynamical properties - instead of directly clustering their activity. Overall the method seems sound and this is an interesting application, but I wish there was some more ground truth analysis on real data. I couldn't find much information about the dataset - what is the rough number of different cell types to be expected? What are other properties of the data? It would be good to compare the results with a simpler method.

**Intersection:**

4: High

**Intersection Comment:**

Develop GLM to include GMM. Use in a model of a neuron.

**Rigor Comment:**

The method seems detailed and convincing - iterate over GLM and GMM. Good description of the generative model as well.

One important comment - it seems like the dataset provided in the link does have 'morphology' and other information about the cell being recorded from. It seems like the authors would definitely want to use this information for a sign that the clusters that they found indeed have something to do with the biology.

**Technical Rigor:**

4: Very convincing

---

### Decision · Program_Chairs · 2019-10-02

Accept (Poster)